# Utonia: Toward One Encoder for All Point Clouds

**Yujia Zhang** [1]  **Xiaoyang Wu** [1]  **Yunhan Yang** [1]  **Xianzhe Fan** [1]  **Han Li** [2]  **Yuechen Zhang** [3]
**Zehao Huang** [2]  **Naiyan Wang** [2]  **Hengshuang Zhao** [1]

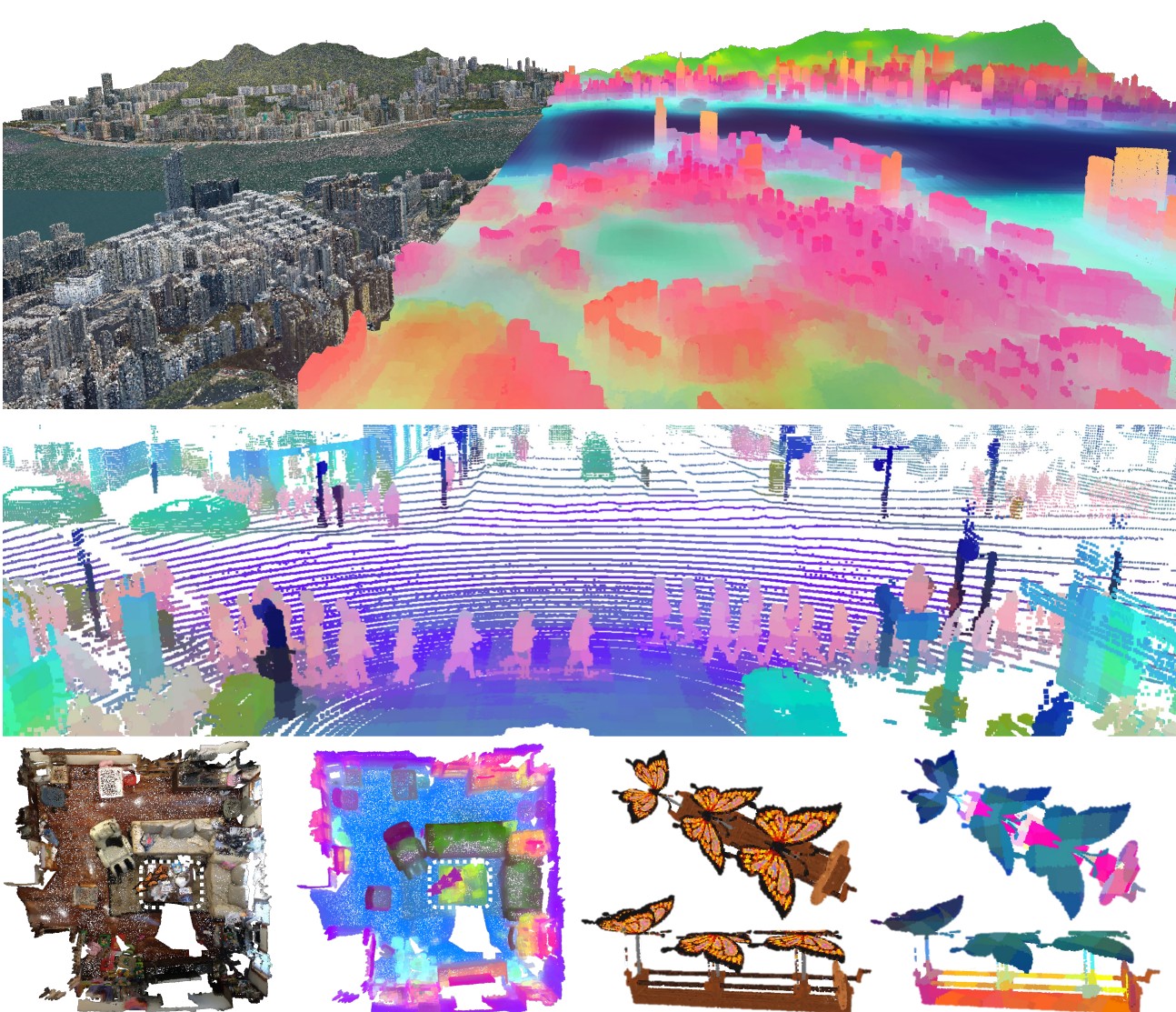

*Figure 1.* **One encoder for all point clouds.** We visualize Utonia features by PCA. City-scale geometry with large coordinate ranges: the smooth, coherent features over terrain and building masses indicate robustness to an extreme extent; Outdoor LiDAR with sparse ring-like scan patterns: features remain organized by geometry rather than scanlines, suggesting reduced reliance on sampling-specific shortcuts. Indoor reconstruction in dense scans: features separate in object level, reflecting geometry-aware scene structure. Object-centric scans with orientation invariance: features exhibit consistent part-level regions with orientation-robust semantics. Despite differences in extent, density, patterns, and coordinate conventions, the representation remains structured and semantically meaningful.

[1]University of Hong Kong [2]Xiaomi [3]The Chinese University of Hong Kong. Correspondence to: Hengshuang Zhao <hszhao@cs.hku.hk>.

*Proceedings of the $43^{rd}$ International Conference on Machine Learning*, Seoul, South Korea. PMLR 306, 2026. Copyright 2026 by the author(s).

## Abstract

We dream of a future where point clouds from all domains can come together to shape a single model that benefits them all. Toward this goal, we present Utonia, a first step toward training a single self-supervised point transformer encoder across diverse domains, spanning remote sensing, outdoor LiDAR, indoor RGB-D sequences, object-centric CAD models, and point clouds lifted from RGB-only videos. Despite their distinct sensing geometries, densities, and priors, Utonia learns a consistent representation space that transfers across domains. This unification improves perception capability while revealing intriguing emergent behaviors that arise only when domains are trained jointly. Beyond perception, we observe that Utonia representations can also benefit embodied and multimodal reasoning: conditioning vision-language-action policies on Utonia features improves robotic manipulation, and integrating them into vision-language models yields gains on spatial reasoning. We hope Utonia can serve as a step toward foundation models for sparse 3D data, and support downstream applications in AR/VR, robotics, and autonomous driving.

## 1. Introduction

Spatial cognition is drawing increasing attention, driven by applications in autonomous driving (Sun et al., 2020; Caesar et al., 2020), AR/VR (Baruch et al., 2021), and robotics (Gu et al., 2023). Today's pipelines remain dominated by dense images and videos (Yang et al., 2025a;b). While effective, these dense formats are redundant carriers of 3D information and do not explicitly enforce geometric consistency. Instead, sparse point clouds provide a geometry-explicit compression of the physical world, yielding a compact representation that complements or conditions dense-centric models.

Yet learning from sparse point clouds is inherently harder than learning from images. Images live on a standard dense grid, whereas point clouds are sparse and unstructured samples. Their scale, density, sampling pattern, and auxiliary modalities vary with sensors and preprocessing, leading to extreme domain shifts. For example, outdoor LiDAR scenes are sparse, span large extents with characteristic scan-line patterns, and often lack color or normal, while object-centric scans are denser and in much smaller ranges. Under such variation, a single pipeline can be dominated by domain-specific priors: scales, sparse patterns, or modality availability, rather than transferable geometry or semantics, causing representations to cluster by domain.

*Table 1.* **Desirable properties of powerful point cloud models:** Sonata scales up point SSL within indoor/outdoor (In./Out.) domain separately. Concerto inherits this recipe and further scales pretraining with video-lifted point clouds. Utonia steps toward one encoder for all these data with additional object assets, supporting optional color/normal input. C./N. indicates how each model handles inconsistent presence of color/normal.

| Models | In. | Out. | Obj. | Video | C./N. |
|---|---|---|---|---|---|
| PointMAE (Pang et al., 2022) | ✗ | ✗ | ✓ | ✗ | w/o |
| PointM2AE (Zhang et al., 2022a) | ✗ | ✗ | ✓ | ✗ | w/o |
| MSC (Wu et al., 2023) | ✓ | ✗ | ✗ | ✗ | w/ |
| Sonata (Wu et al., 2025) | ✓₁ | ✓₂ | ✗ | ✗ | w/ |
| Concerto (Zhang et al., 2025b) | ✓₁ | ✓₂ | ✗ | ✓₁ | w/ |
| Utonia | ✓ | ✓ | ✓ | ✓ | adaptive |

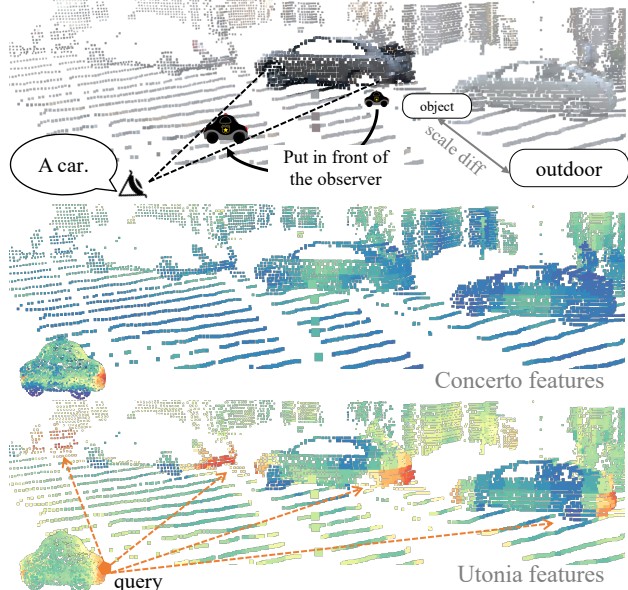

*Figure 2.* **Cross-domain semantic similarity.** Human perception operates at a fixed angular resolution, resulting in similar perception granularity between a close small toy car and a far-away real car, which motivates semantic matching at a canonical granularity across domains. Utonia representations exhibit high similarity between the toy car from object CAD and real cars in outdoor scenes, while the previous SOTA Concerto fails to align.

Thus, unlike dense images, universality does not arise in point cloud self-supervised learning (SSL) by simply mixing datasets. Point cloud SSL remains domain-fragmented due to practical design choices rather than the structure of the physical world. Specifically, prior methods, Sonata (Wu et al., 2025) and Concerto (Zhang et al., 2025b), are typically trained within a single domain and transfer poorly across domains, even though all point clouds are discrete observations of the same 3D reality. As shown in Figure 2, when querying the front of a toy car from the object CAD and retrieving it by feature similarity in an outdoor LiDAR scene, Concerto struggles to highlight the corresponding region on the real car.

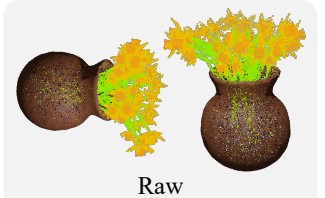 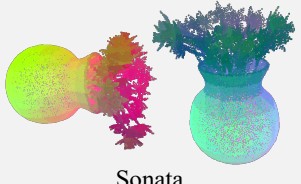 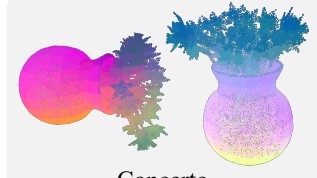 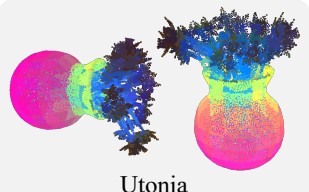

| Raw | Sonata | Concerto | Utonia |

*Figure 3.* **Gravity priors influence.** Scene-level data have a strong z-axis up prior. Utonia steps further to erase such assumptions by including rotation-invariant objects with strong SE(3) augmentations into pretraining datasets.

However, recent progress has matured both ingredients needed for unification: scalable point backbones and robust SSL recipes. PTv3 (Wu et al., 2024a) offers a strong foundation, while Sonata (Wu et al., 2025) and Concerto (Zhang et al., 2025b) mitigate geometric shortcuts via perturbations and cross-modal synergy, making large-scale training stable within individual domains. We argue the next step is to move beyond siloed pretraining and learn a single encoder across domains. *It is time to turn fragmented 3D observations into a shared representation.*

Motivated by this, we take a first step toward an *all-for-one, one-for-all* point encoder: learned *from* and shared *for* diverse point clouds. We present **Utonia**, not as a definitive solution, but as an exploration that exposes what breaks joint training and distills a minimal set of fixes. Specifically, we identify three core mismatches across domains: non-unified input channels (e.g., optional color/normal/image cues), inconsistent sparsity and density, and perceptual granularity shifts induced by scenes and downstreams. Utonia addresses them with three simple, domain-agnostic designs: *Causal Blinding* via randomized modality masking (e.g., dropping color/normal) for robustness to missing channels, *granularity-prompted rescaling* to harmonize effective perception scales, and *RoPE-enhanced positional hints* for more transferable geometry encoding, while avoiding domain-specific modules. With these minimal fixes, we can stably train a single encoder on 250k cross-domain point clouds, augmented with 1M additional CAD assets, yielding a unified representation space and our cross-domain pretrained model, Utonia.

Surprisingly, joint pretraining yields several emergent behaviors. First, object, indoor, and outdoor data begin to benefit under a shared encoder, rather than compete across domains. Second, the learned representations retain scene-level gravity alignment while remaining largely gravity-irrelevant for object-centric geometry. Beyond 3D perception, the pretrained encoder benefits spatial reasoning when integrated into VLM backends and improves robotic manipulation when conditioning VLA policies. We hope these highlight sparse point clouds as a compelling substrate toward unified spatial cognition, turning fragmented 3D observations into a shared representation. *Project page:* https://pointcept.github.io/Utonia

*Table 2.* **Scale and grid size.** Naive joint pretraining with per-dataset default grid sizes is unstable and causes large performance drops. Grid size jittering offers limited robustness. Fixing a global grid size and rescaling coordinates to a consistent granularity substantially improves cross-domain joint training and transfer.

| Scale & Grid Size | ScanNet200 Val | | | Waymo Val | | | PartNetE |
|---|---|---|---|---|---|---|---|
| Methods | mIoU | mAcc | allAcc | mIoU | mAcc | allAcc | mIoU |
| Separate domain | 34.4 | 45.7 | 82.9 | 60.5 | 71.6 | 88.9 | 41.6 |
| Origin grid size | 29.1 | 39.2 | 80.7 | 43.9 | 56.8 | 83.6 | 39.5 |
| Jitter grid size | 29.6 | 40.3 | 80.7 | 42.6 | 55.8 | 82.7 | 38.7 |
| Fixed grid size | **33.5** | **46.1** | **82.0** | **56.6** | **69.1** | **87.9** | **43.5** |

## 2. Pilot Study: What Prevents Unification?

A naive way to scale sparse 3D pretraining to multiple domains is to simply merge all datasets and train one encoder jointly. Empirically, this direct joint training is often suboptimal and unstable as in Tab. 2. To understand the underlying causes, we conduct a pilot study and summarize three failure symptoms in practice:

**Sensitivity to granularity shifts.** In sparse point processing, the grid size defines the metric unit of local neighborhoods: the same architectural operator can cover centimeters in one domain but meters in another. Such granularity shifts change neighborhood statistics and local topology, altering how points are grouped and interact. Thereby, the learned features are coupled to the domain-specific scale. As shown in Tab. 2, while per-domain optimal settings work well in separate training, directly merging domains makes joint training highly sensitive: using domain-specific original grids or jittering them leads to drops. These observations motivate a principled alignment of spatial units for joint pretraining. Inspired by the intuition that an observer operates with a roughly fixed minimal resolution, we define a standard observing granularity and rescale each point cloud accordingly, analogous to changing the observation distance in Fig. 2, so that positional hints and interactions can be built upon comparable spatial units across domains. This granularity alignment stabilizes joint training and improves cross-domain transfer.

**Bias toward gravity convention.** Many scene-level point clouds with a coarser granularity follow a gravity-aligned convention, making height a physical reference. Accordingly, prior scene-centric training usually avoids strong $x/y$ rotations, since many semantics depend on gravity, such as

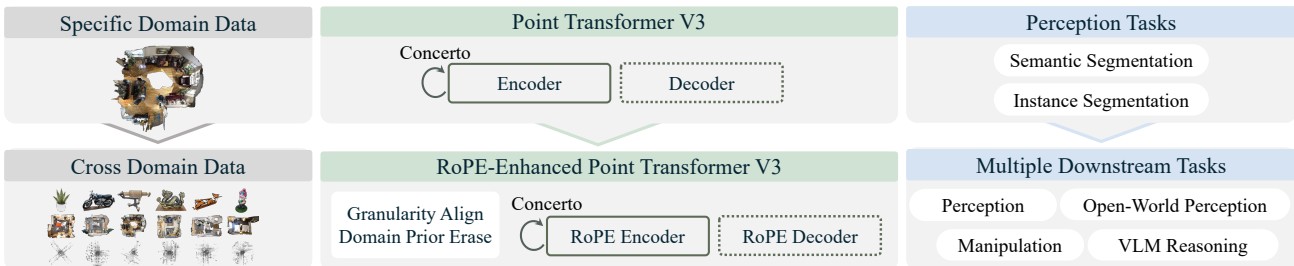

*Figure 4.* **Overview of Utonia.** Utonia introduces three critical improvements to the point cloud SSL pipeline. Cross-domain data: jointly training on object-centric, indoor, and outdoor point clouds. RoPE-Enhanced Point Transformer V3: Strengthening spatial encoding and cross-domain transferability via RoPE on granularity-aligned coordinates and domain-prior erasure. Broader evaluation: extending beyond standard perception tasks to spatial reasoning, robotic manipulation, and open-world part segmentation.

floor vs. ceiling, and supporting surfaces. However, height can act as a domain cue, which hurts transfer to object-centric scans with a fine granularity. In Fig. 3, features from Sonata and Concerto exhibit strong $z$-correlated patterns, while Utonia weakens this correlation. This suggests treating gravity alignment as a granularity-dependent prior: retaining upright structure for scene-scale scans while encouraging rotation invariance for fine-grained objects.

**Inconsistent modality availability.** Point clouds across domains expose different auxiliary channels beyond coordinates: colors and normals. In naive multi-domain pre-training, the encoder tends to exploit these channels whenever they exist, since they provide strong cues that can complement coordinates. However, the resulting dependence is unstable: when these modalities are missing, noisy, or defined differently, representations and performance can degrade, as shown in Tab. 7. Intuitively, this resembles a person who can walk well with vision but stumbles when suddenly blindfolded. This motivates a specific design on modality availability during pretraining, enabling the model to benefit from optional modalities when present while remaining robust when they are absent.

**Conclusion.** The above failure modes reveal two dominant sources of multi-domain pretraining. First, *spatial coordinate discrepancies* include sensitivity to discretization granularity and bias toward a gravity-aligned convention. Second is *inconsistent modality availability*. If these are not properly addressed, joint training can be dominated by domain-specific cues, resulting in a degraded overall performance and weaker cross-domain transfer.

## 3. Utonia

This section presents the methodology of Utonia, which aims to pretrain a single Point Transformer V3 (Wu et al., 2024a) encoder across various domains: indoor scans, outdoor LiDAR, remote sensing, object CAD, and video-lifted point clouds. Detailed datasets and training settings are provided in B.1. We follow the teacher-student self-distillation pretraining recipe established in (Wu et al.,

2025; Zhang et al., 2025b) and focus on the designs that facilitate multi-domain training, which is hindered by the previously mentioned two reasons: To address varying modality availability, we introduce Causal Modality Blinding in Sec. 3.1, and further harmonize spatial coordinate discrepancies with a two-step design: Perceptual Granularity Rescale in Sec. 3.2, and RoPE on Granularity-Aligned Coordinates in Sec. 3.3.

### 3.1. Causal Modality Blinding

Point clouds across domains do not share a standardized input form. Some provide auxiliary channels beyond coordinates, while others do not. To make joint pretraining feasible, we first define a unified modality interface as the default setting: we concatenate coordinates, colors, and normals, using default zeros for missing modalities. While this unified interface allows a single encoder to take all domains as input, it also exposes a major source of noise and shortcut learning: under naive multi-domain pretraining, the encoder tends to exploit the richest available channels whenever they exist, and modality statistics can become domain-identifying cues, leading to unstable transfer when modalities are absent or shift in distribution as introduced in Sec. 2. We therefore introduce Causal Modality Blinding to intervene on modality availability during pretraining explicitly. Concretely, we treat each modality group as optional except for coordinates and apply blinding at two levels: per-data blinding randomly drops entire modality groups for a sample, and per-point blinding further masks modalities at individual points. Intuitively, this is analogous to training a person under deliberate vision deprivation: by practicing "blindfolded" during learning, the model remains functional when color or normal cues are absent while still leveraging them when available.

### 3.2. Perceptual Granularity Rescale

Point clouds are discrete samples of the same continuous 3D world, yet their spatial extent, sampling patterns, and coordinate conventions vary widely across sensors and do-

*Table 3.* **Indoor semantic segmentation.** Utonia shows competitive mIoU results compared with Concerto across indoor benchmarks.

| Indoor Sem. Seg. | Params | | ScanNet Val | | | ScanNet200 Val | | | ScanNet++ Val | | | S3DIS Area 5 | | |
|---|---|---|---|---|---|---|---|---|---|---|---|---|---|---|
| Methods | Learn. | Pct. | mIoU | mAcc | allAcc | mIoU | mAcc | allAcc | mIoU | mAcc | allAcc | mIoU | mAcc | allAcc |
| ○ PTv3 (Wu et al., 2024a) | 124.8M | 100% | 77.6 | 85.0 | 92.0 | 35.3 | 46.0 | 83.4 | 48.2 | 61.6 | 87.0 | 73.4 | 78.9 | 91.7 |
| ● Sonata (Wu et al., 2025) (lin.) | <0.2M | <0.2% | 72.5 | 83.1 | 89.7 | 29.3 | 41.6 | 81.2 | 38.9 | 52.8 | 84.3 | 72.3 | 81.2 | 90.9 |
| ● Concerto (Zhang et al., 2025b) (lin.) | <0.2M | <0.2% | 77.3 | 86.6 | 91.7 | **37.4** | 49.5 | 83.3 | **45.6** | 60.5 | 86.5 | 73.5 | **81.3** | 90.9 |
| ● Utonia (lin.) | <0.2M | <0.2% | **77.7** | **87.0** | **91.9** | 36.4 | **49.8** | **83.4** | 44.7 | 58.4 | 86.3 | **74.7** | 81.2 | **91.5** |
| ● Sonata (Wu et al., 2025) (dec.) | 16.3M | 13% | 79.1 | 86.6 | 92.7 | 33.5 | 44.5 | 84.1 | 45.2 | 57.4 | 86.8 | 74.5 | 80.4 | **92.6** |
| ● Concerto (Zhang et al., 2025b) (dec.) | 16.3M | 13% | 79.5 | 87.6 | 92.6 | 37.8 | **50.5** | 84.1 | 48.3 | **62.3** | **87.7** | 75.5 | **84.2** | 92.3 |
| ● Utonia (dec.) | 20.5M | 13% | **80.3** | **88.3** | **92.9** | **38.0** | 49.9 | **84.9** | **48.5** | 58.8 | 87.3 | **76.2** | 82.0 | 92.4 |
| ● MSC (Wu et al., 2023) | 124.8M | 100% | 78.2 | 85.3 | 92.2 | 33.4 | 43.7 | 83.4 | 42.4 | 53.6 | 85.9 | 69.9 | 74.9 | 91.2 |
| ● PPT (Wu et al., 2024b) (sup.) | 124.8M | 100% | 78.6 | 85.9 | 92.3 | 36.0 | 46.2 | 83.8 | 43.3 | 55.7 | 86.4 | 74.3 | 80.1 | 92.0 |
| ● Sonata (Wu et al., 2025) (f.t.) | 124.8M | 100% | 79.4 | 86.1 | 92.5 | 36.8 | 46.5 | 84.4 | 43.7 | 55.8 | 86.6 | 76.0 | 81.6 | 93.0 |
| ● Concerto (Zhang et al., 2025b) (f.t.) | 124.8M | 100% | 80.7 | 87.4 | 93.1 | 39.2 | 50.2 | 85.0 | **50.7** | **63.3** | **87.9** | 77.4 | 85.0 | 93.2 |
| ● Utonia (f.t.) | 157.7M | 100% | **81.1** | **89.1** | **93.3** | **39.6** | **51.4** | **85.1** | 49.0 | 59.2 | 87.2 | **78.1** | **86.8** | **93.4** |

*Table 4.* **Outdoor semantic segmentation.** We validate Utonia's better performance on various outdoor segmentation benchmarks than Concerto. Noted that the results of Sonata are PPT supervised f.t. with additional datasets, while Utonia is directly f.t..

| Outdoor Sem. Seg. | Params | | NuScenes Val | | | Waymo Val | | | SemanticKITTI Val | | |
|---|---|---|---|---|---|---|---|---|---|---|---|
| Methods | Learn. | Pct. | mIoU | mAcc | allAcc | mIoU | mAcc | allAcc | mIoU | mAcc | allAcc |
| ○ PTv3 (Wu et al., 2024a) | 124.8M | 100% | 80.4 | 87.2 | 94.7 | 71.3 | 80.5 | 94.7 | 70.8 | 76.1 | 92.6 |
| ● Sonata (Wu et al., 2025) (lin.) | <0.2M | <0.2% | 66.1 | 77.2 | 92.4 | 60.5 | 72.5 | **92.5** | 62.0 | 72.5 | 91.0 |
| ● Concerto (Zhang et al., 2025b) (lin.) | <0.2M | <0.2% | 74.2 | 83.2 | 93.3 | 62.2 | 74.4 | 89.4 | 66.6 | 76.2 | 91.4 |
| ● Utonia (lin.) | <0.2M | <0.2% | **75.5** | **84.8** | **93.7** | **63.8** | **75.4** | 89.9 | **67.7** | **76.9** | **91.9** |
| ● Sonata (Wu et al., 2025) (dec.) | 16.3M | 13% | 77.3 | 85.9 | 94.2 | 69.1 | 78.8 | **94.3** | 68.4 | **76.5** | 92.3 |
| ● Concerto (Zhang et al., 2025b) (dec.) | 20.5M | 13% | 77.5 | 86.5 | 94.1 | 69.4 | **79.8** | 91.1 | 69.3 | 76.2 | 92.3 |
| ● Utonia (dec.) | 20.5M | 13% | **78.2** | **87.1** | **94.3** | **69.4** | 79.7 | 91.4 | **70.0** | 75.9 | **92.7** |
| ● Sonata (Wu et al., 2025) (f.t.) | 124.8M | 100% | 81.7 | 87.9 | **95.0** | **72.9** | **81.9** | **94.9** | **72.5** | **77.9** | 93.2 |
| ● Concerto (Zhang et al., 2025b) (f.t.) | 157.7M | 100% | 82.0 | 88.1 | 94.6 | 69.2 | 77.8 | 91.6 | 71.2 | 77.0 | 92.5 |
| ● Utonia (f.t.) | 157.7M | 100% | **82.2** | **88.3** | 94.8 | 71.4 | 80.0 | 91.8 | 72.0 | 77.8 | **93.5** |

mains. As a result, the same architectural unit can correspond to vastly different metric extents across datasets. This scale mismatch encodes domain-specific geometry into local operators and positional hints, making it difficult to share geometric patterns in joint training. Inspired by the idea that an observer operates with a roughly fixed minimal angular resolution, we rescale coordinates so that each point cloud matches a shared perceptual granularity before positional encoding. This maps inputs from vastly different extents into a comparable coordinate space for joint training, but it does not force all domains to share the same coordinate convention. Scene-scale scans with a coarser granularity are often gravity-aligned, where height encodes stable physical relations, while object-scale scans with a fine granularity should appear in arbitrary orientations. We treat gravity as a prior that depends on the original desired granularity: we preserve upright structure for scenes, and encourage orientation invariance for objects.

We draw the rescaling factor from a range based on the original desired granularity, thereby exposing Utonia to different effective granularities during pretraining. Through asymmetric scales, rotations, and shifts in teacher and student views, we enforce cross-view feature consistency in self-supervised learning. At inference, the rescaling factor can be adjusted to match the desired granularity.

### 3.3. RoPE Bridges Granularity-Aligned Coordinates

Rescaling aligns the basic spatial unit across domains, but it does not specify how relative geometry should modulate token interactions. Sparse convolution encodes position through discretization, thus coupling interactions to discretization, local neighborhoods, and density nonuniformity. Driven by the hypothesis that a continuous embedding can provide a unified positional hint, we surprisingly found that adopting RoPE on granularity-aligned coordinates perfectly achieves this and benefits cross-domain learning. RoPE provides a parameter-free positional encoding applied directly to attention queries and keys, naturally supporting extendable point sets with different densities. Combined with our perceptual rescaling and coordinate perturbations, RoPE helps attention rely on local geometry rather than memorizing domain-specific coordinate conventions. This is particularly beneficial under density variation: outdoor LiDAR is dense at close range and sparse at far range, and cross-domain sampling patterns further cause neighborhood structure to fluctuate. RoPE does not alter neighborhood construction, but it encourages attention toward continuous relative geometry, making it less

*Table 5.* **Object classification and part segmentation.** We adopt object classification tasks, ModelNet40 and ScanObjectNN, and object part segmentation tasks, ShapeNetPart and PartNetE, to evaluate Utonia, showing competitive performance with Concerto. For ScanObjectNN, we report the results of OBJ-BG and PB-T50-RS (H subscript).

| Object Cls. & Part Seg. | Params | | ModelNet40 | | ScanObjectNN | | ScanObjectNN(H) | | ShapeNetPart | | PartNetE |
|---|---|---|---|---|---|---|---|---|---|---|---|
| Methods | Learn. | Pct. | mAcc | allAcc | mAcc | allAcc | mAcc | allAcc | i.mIoU | c. mIoU | mIoU |
| ○ PTv3 (Wu et al., 2024a) | 124.8M | 100% | 90.1 | 92.9 | 87.5 | 89.5 | 80.0 | 82.3 | 81.9 | 80.2 | 12.7 |
| ● Sonata (Wu et al., 2025) (lin.) | <0.2M | <0.2% | 85.1 | 89.8 | 80.3 | 82.8 | 75.2 | 76.2 | 83.6 | **81.6** | 52.0 |
| ● Concerto (Zhang et al., 2025b) (lin.) | <0.2M | <0.2% | 88.2 | 90.7 | 86.1 | **87.4** | 78.5 | 79.7 | **83.9** | 81.5 | **55.8** |
| ● Utonia (lin.) | <0.2M | <0.2% | **88.2** | **90.8** | **86.9** | 87.3 | **78.8** | **80.4** | 82.5 | 80.6 | 39.8 |
| ● Point-M2AE (Zhang et al., 2022a) (f.t.) | 12.9M | 100% | / | 94.0 | / | 91.2 | / | 86.4 | **86.5** | **84.9** | / |
| ● Sonata (Wu et al., 2025) (f.t.) | 124.8M | 100% | 92.1 | 94.1 | 92.2 | 92.8 | 85.4 | 87.5 | 84.4 | 83.6 | 61.6 |
| ● Concerto (Zhang et al., 2025b) (f.t.) | 137.4M | 100% | 92.4 | 94.1 | 92.3 | 92.9 | 86.7 | 88.1 | 86.1 | 83.6 | 60.8 |
| ● Utonia (f.t.) | 137.4M | 100% | **92.4** | **94.3** | **95.0** | **95.2** | **88.3** | **89.9** | 86.3 | 84.2 | **62.7** |

sensitive to such sampling-induced changes.

Additionally, following DINOv3 (Siméoni et al., 2025), we apply anisotropic coordinate jittering and isotropic scaling to canonical coordinates for augmented coordinates. We compute RoPE from the augmented coordinates and apply it in every attention layer to rotate queries and keys. This discourages the model from binding semantics to a particular unit convention or axis scaling, improving robustness under cross-domain shifts. Details are in B.2.

# 4. Experiments

We benchmark Utonia across indoor, outdoor, and object-centric 3D tasks under the standard evaluation protocols of Sonata and Concerto. For indoor and outdoor semantic segmentation, we report all linear probing, decoder probing, and full fine-tuning. For object benchmarks, we report linear probing and full fine-tuning. For object classification, we fine-tune end-to-end with a classification head without a multi-scale decoder. To enable fair cross-domain comparisons, we also retrain prior baselines on domains they do not originally report: Sonata on objects; Concerto on outdoor and objects. All our full fine-tuning implementations are trained directly on the target datasets without PPT (Wu et al., 2024b), while the results from the Sonata paper, indoor and outdoor results, are supervised by PPT. Object classification is reported without voting. Finally, we analyze behaviors that emerge under cross-domain joint pretraining and validate Utonia on multiple downstream tasks.

**Indoor semantic segmentation.** In Tab. 3, Utonia achieves strong indoor segmentation performance. With full fine-tuning, it reaches SOTA mIoU on multiple benchmarks (e.g., 81.1% on ScanNet and 78.1% on S3DIS), and slightly outperforms Concerto under decoder probing. On the finer-grained ScanNet200 and ScanNet++, Utonia trails Concerto under linear probing, but the gap largely closes with decoder probing, suggesting that Utonia benefits from a non-linear spatial head beyond a single linear layer.

**Outdoor semantic segmentation.** In Tab. 4, Utonia

achieves the best mIoU among compared methods under linear and decoder probing, and slightly surpasses Concerto under full fine-tuning. For reference, we include Sonata results from the original paper. These numbers use additional supervised tuning with PPT (Wu et al., 2024b) and extra data, and are therefore not directly comparable to the purely target-dataset fine-tuning setting used for Utonia.

**Object part segmentation and classification.** We evaluate Utonia on object-centric classification and part segmentation in Tab. 5. Utonia shows strong linear-probing transfer on classification, while its gains on part segmentation appear under full fine-tuning with a task decoder. The drop in linear probing and rise from full fine-tuning of part segmentation suggest that the pretrained representation captures object-level semantics well, while fine-grained part understanding benefits from end-to-end adaptation and becomes less linearly decodable. This motivates a stronger spatial head to query part-aware signals, or a set of lightweight register tokens to store object-level information with part-level structure accessible via a linear probe.

**Colors/Normals missing.** Tab. 7 validates Utonia's robustness on ScanNet and NuScenes when colors and normals are missed or unreliable. For object-centric data, ScanNetObjectNN and ModelNet40 do not provide color information, the results of which can be found in Tab. 5. Concerto performs well when both color and normals are available, but degrades substantially when the modality is absent, especially colors. In contrast, Utonia remains more stable under missing color/normal inputs, suggesting that our modality-dropout training discourages over-reliance on modality availability and improves robustness when colors or normals may be unavailable or unreliable. We also find that removing normals typically causes a smaller drop than removing color, possibly because normals in large-scale data can be noisy or inconsistently defined, and coarse orientation cues can often be inferred from local geometry.

*Table 6.* **Ablation study.** The default ablation setup trains on ScanNet, Structured3D, PartNet, and Waymo with 38M PTv3 model. All of our designs are enabled by default. Default settings are in `blue`. If not specified, the results are from linear probing.

*(a)* **Object augmentation.** Larger augmentations benefit objects.

| Object Scale | ScanNet200 | Waymo | PartNetE | ScanObj.(H) |
|---|---|---|---|---|
| baseline | **34.9** | 59.2 | 44.6 | **66.9** |
| less scale aug. | 33.7 | 59.7 | 43.0 | 64.5 |
| less rot. aug. | 34.3 | **60.8** | **47.5** | 63.0 |

*(b)* **RoPE.** RoPE effects across domains.

| RoPE | Single-Data | + RoPE | Multi-Data | + RoPE |
|---|---|---|---|---|
| ScanNet200 | 34.4 | 34.0 (-0.4) | 33.5 | **34.9** (+1.4) |
| Waymo | 60.5 | **62.1** (+1.6) | 56.6 | 59.2 (+2.6) |
| PartNetE | 44.1 | 42.4 (-1.7) | 43.5 | **44.6** (+1.1) |

*(c)* **RoPE base choice.** Utonia is robust to different RoPE bases.

| RoPE Base | ScanNet200 | Waymo | PartNetE |
|---|---|---|---|
| w/o RoPE | 33.5 | 56.6 | 43.5 |
| 1 | 34.0 | **60.6** | 43.9 |
| 10 | **34.9** | 59.2 | 44.6 |
| 100 | 34.1 | 59.2 | **45.0** |
| 1000 | 34.9 | 60.1 | 43.7 |

*(d)* **Modality blinding.** Randomly drop per sample and per point.

| Color Blinding | ScanNet200 | | Waymo | | PartNetE | |
|---|---|---|---|---|---|---|
| dropping strategy | w/ c. | w/o c. | w/ c. | w/o c. | w/ c. | w/o c. |
| w/o drop | 34.7 | 9.3 | 56.5 | 54.5 | 43.5 | **44.3** |
| drop at loading | **34.9** | **31.7** | 59.2 | **58.1** | **44.6** | 39.8 |
| drop at masked views | 34.1 | 10.1 | 56.3 | 54.4 | 42.1 | 41.1 |
| drop at local views | 34.5 | 29.4 | **59.6** | 58.1 | 43.4 | 37.6 |

*(e)* **Augmentations.** Different augmentations help generalization.

| Augmentation | ScanNet200 | Waymo | PartNetE |
|---|---|---|---|
| baseline | 34.9 | 59.2 | **44.6** |
| w/o frame aug. | 34.1 | 58.8 | 40.0 |
| w/o RoPE aug. | **35.0** | **60.1** | 44.1 |
| w/o scale aug. | 34.3 | 60.0 | 42.6 |

*(f)* **Scale up.** Data and model scale are crucial for crossing domains.

| Scale Up | ScanNet200 | | Waymo | | PartNetE | |
|---|---|---|---|---|---|---|
| scale | lin. | ft. | lin. | ft. | lin. | ft. |
| 38M with 83k data | 34.9 | 36.9 | 59.2 | 68.5 | 44.6 | 53.2 |
| 137M with 83k data | 36.0 | 38.7 | 62.6 | 70.9 | **45.6** | 55.7 |
| 137M with all data | **36.4** | **39.6** | **63.8** | **71.4** | 39.8 | **62.7** |

## 4.1. Ablations

The ablation setting is provided in the caption of Tab. 6. Notably, the channels for RoPE-enhanced Point Transformer V3 should be divisible by 6. The architecture settings for PTv3 in Utonia are provided in Tab. 9.

**Scale and grid size.** In Tab. 2, domain-specific grid sizes hinder multi-domain pretraining, and grid-size jittering offers limited benefit. A unified grid size with coordinate rescaling to a consistent granularity stabilizes training and improves cross-domain transfer.

**Stronger scale augmentation for objects.** Objects are usually in a normalized coordinate system, which is quite different from indoor and outdoor settings. We apply much larger scaling parameters than scene-level data. For scene-level data, we only jitter the scale at $\pm10\%$, while we improve this to $\pm50\%$ for objects. We use the default ablation setting and evaluate additionally on out-of-domain ScanObjectNN PB-T50-RS to validate the effect. As in Tab. 6a, applying the same level scale augmentation as indoor data to object data leads to a performance drop on both the part segmentation task on PartNetE and the classification task on ScanObjectNN.

**Stronger rotation augmentation for objects.** To reduce reliance on the $z$ axis without disrupting gravity-aligned scene understanding, we include rotation-invariant object data in pretraining datasets and apply a stronger rotation transformation to them. For scene-level data, we use full yaw augmentation, $z$-axis rotation in $[-\pi, \pi]$, together with only mild roll/pitch perturbations, $x/y$ rotations in $\left[-\frac{\pi}{64}, \frac{\pi}{64}\right]$. For object-centric data, where orientation is in-

herently arbitrary, we apply full SO(3) rotations by sampling independent rotations for all three axes in $[-\pi, \pi]$. Tab. 6a shows that reducing object rotations to the mild scene-level setting significantly degrades performance on ScanObjectNN PB-T50-RS, which contains strong rotation, scale, and shift perturbations. As for PartNetE, it is evaluated in strong gravity alignment, like scene-level data. So applying strong SO(3) rotations can hurt its evaluation performance. Thus, the results on PartNetE cannot reflect the performance of gravity-align erasing.

**RoPE effects across domains.** In multi-domain training, geometric interactions better remain compatible across domains with different densities, scales, and coordinate conventions; otherwise, local neighborhoods and token interactions become hard to share. In Tab. 6b, RoPE consistently improves the multi-domain performance across ScanNet200, Waymo, and PartNetE, with the most pronounced gain on the Waymo benchmark. Notably, this improvement also appears under single-domain training for Waymo because outdoor LiDAR exhibits severe density non-uniformity even within a single scan. By providing a continuous relative-geometry positional cue, RoPE makes attention less sensitive to density non-uniformity, which also improves robustness to cross-domain density shifts and yields consistent gains in multi-domain training.

**RoPE base choice.** The RoPE base $B$ controls the frequency scale of rotary positional encoding. Tab. 6c shows that performance is stable across a wide range of $B$. Tiny bases make the phase rotate too rapidly for large coordinate values, causing phase wrapping. However, in PTv3, this is acceptable to some extent because the local attention relies

on neighborhood geometry rather than global positioning. We set $B=10$ as the default.

**Modality blinding.** We ablate the place to apply modality blinding during pretraining: at data loading, on local views, or on masked views. Tab. 6d shows that dropout at loading yields the strongest robustness. As all objectives are consistently trained under modality-missing inputs, the encoder cannot treat modality availability as a shortcut. Dropping only on local views is also effective: the student learns to handle missing modalities while receiving guidance from the teacher's complete views. In contrast, applying dropout on masked views is least effective, because masked views are already information-limited, removing color or normal can over-corrupt the input.

**Frame augmentation.** For dynamic data, we construct temporal views by feeding the teacher a pose-aligned multi-frame aggregated global point cloud while randomly sampling a single-frame local point cloud for the student. In Tab. 6e, temporal frame augmentation benefits sequence data Waymo, as aggregating pose-aligned frames densifies point clouds and mitigates scan-pattern artifacts.

**RoPE coordinate augmentation.** We also explore perturbing the coordinates used by RoPE without changing the input geometry to promote scale invariance. This encourages the encoder to treat the input point cloud as a region under varying scales across iterations, effectively providing an implicit scale augmentation without explicitly constructing multi-scale inputs. As in Tab. 6e, though this perturbation does not show significant improvement, we enable it by default to improve robustness to various scales and anticipate it will be more beneficial as pretraining scales become broader and noisier across domains.

**Random scale augmentation.** We apply random scaling before grid sampling to improve robustness to scale variation, which enables flexible inference under different scales and normalization conventions. As shown in Tab. 6e, removing this augmentation slightly degrades PartNetE performance, so we keep it enabled by default. For normalized object-centric data, we use a wider scale range.

**Scale up.** We scale along both model capacity and data size. We first increase the backbone from 38M to 137M under the same 83k multi-domain data mixture, and then scale the pretraining data to the full mixture, 250k data plus 1M sampled Cap3D objects. In Tab. 6b, 37M PTv3 with RoPE under multi-domain pretraining achieves performance comparable to training separate domain-specific models on indoor and object benchmarks, while showing a slight drop on the outdoor benchmark. Here, when scaling up the backbone in Tab. 6f, multi-domain pretraining consistently surpasses domain-specific pretraining overall, suggesting that the smaller model is capacity-limited under

cross-domain training. Further scaling the data brings continued gains on indoor and outdoor benchmarks. On Part-NetE, scaling data improves fine-tuning but reduces linear probing, suggesting that the representation contains both instance-level semantics and part-aware cues, but they are not equally accessible under a fixed linear readout. In particular, larger and noisier mixtures may favor transferable object-level semantics, while fine-grained part cues are better recovered through task-specific adaptation. These observations highlight different readout requirements going forward: global registers or a [CLS]-like token for classification, and a query-based decoder for part segmentation.

### 4.2. Downstream Applications

**Robotics manipulation.** We report the manipulation test success rate of Sonata, Concerto, and Utonia in Tab. 11, showing Utonia's advance in object-centric robotics tasks using the setting in Any3D-VLA (Fan et al., 2026). The detailed experiment settings are in B.3. Fig. 5 shows that Utonia features remain coherent in cluttered RGB-D (Khazatsky et al., 2024). Support surfaces form consistent regions, while object areas exhibit distinct structure under heavy occlusion and partial observations. These qualitative results show the generalization of manipulation scenes.

**Open-world object segmentation.** We further evaluate Utonia on open-world 3D object segmentation by building on $P^3SAM$ (Ma et al., 2025). Fig. 6 compares Sonata and Utonia under the same promptable segmentation setting described in B.3 with the detailed quantitative segmentation results in Tab. 10. The Sonata-initialized encoder produces features lacking distinct part semantics, resulting in poor segmentation quality. In contrast, the Utonia-initialized encoder generates features with clear part-level structures, enabling highly accurate segmentation with well-defined boundaries and consistent semantic meaning. These results suggest that unified multi-domain pretraining strengthens part-level representations for open-world 3D segmentation.

**Spatial reasoning.** We evaluate Sonata, Concerto, and Utonia using Video-3D LLM (Zheng et al., 2025) on 3 representative tasks: 3D visual grounding, 3D dense captioning, and 3D question answering with details in B.3. Tab. 12 shows that Utonia yields consistent gains on grounding and question answering, while remaining competitive on dense captioning. These results suggest that a unified point encoder provides geometry-aware cues when fused into dense video-based vision language models, improving 3D scene understanding beyond single-domain pretraining.

## 5. Related Works

**Point self-supervised learning.** Self-supervised representation learning for point clouds has been extensively stud-

ied, but most prior works are developed within a single domain: Afham et al. (2022); Rao et al. (2020); Sauder & Sievers (2019); Pang et al. (2022); Zhang et al. (2022a); Qi et al. (2023) focus on object-level representations. Huang et al. (2021); Wang et al. (2021); Xie et al. (2020) dive into indoor scene applications besides objects. Zhang et al. (2021) extends the indoor scenes to outdoor point clouds. Tian et al. (2023) focuses on outdoor geometry representation learning. These can be categorized into two aspects: reconstruction and contrastive learning. Recently, Sonata (Wu et al., 2025) proposed a reliable form to handle the geometry shortcuts. Concerto (Zhang et al., 2025b) further incorporates cross-modal prediction together with point SSL. Despite this progress, learning a single point cloud encoder that generalizes across domains remains challenging due to large shifts in extent, density, sampling patterns, and modality availability.

**Unified models in point clouds.** Only a limited number of works explicitly target unified point cloud pretraining across various point-cloud types. Zhang et al. (2021) focuses on adapting the models for different inputs like voxels and points, and generalizes to outdoor situations by pretraining on outdoor data from scratch. Zha et al. (2025) proposed the Mixture-of-Domain-Experts Model to adapt to data across domains. Zhang et al. (2022b) splits a large point cloud into occupied volumes to deal with different densities. While promising, existing solutions often rely on complex additional modules or are validated on a limited subset of domain combinations. In contrast, we pursue a unified paradigm that shapes one encoder trained jointly on indoor, outdoor, and object-centric point clouds.

**3D Rotary Position Embedding.** Rotary positional embedding (RoPE) was introduced first in (Su et al., 2024) and has become a popular positional encoding in 1D large language models (Touvron et al., 2023) and 2D vision models (Heo et al., 2024; Siméoni et al., 2025). In 3D point transformers, earlier designs adopt relative positional embeddings (Zhao et al., 2021; Wu et al., 2022) or conditional position encoding by sparse convolution (Wu et al., 2024a). Recently, LitePT (Yue et al., 2025) leverages RoPE in the backbone for an efficiency tradeoff. Our focus is different: we study how RoPE, together with granularity-aligned coordinates, improves cross-domain transfer in large-scale multi-domain point cloud SSL.

## 6. Conclusion and Future

We present Utonia, a first step toward an *all-for-one, one-for-all* point encoder learned from diverse point clouds and shared for diverse point clouds. By identifying cross-domain mismatches and addressing them with minimal, domain-agnostic designs, Utonia enables joint pretraining and yields a shared representation space with emergent cross-domain benefits. Beyond 3D perception, Utonia's features transfer to spatial reasoning in VLM backends and improve robotic manipulation when conditioning VLA policies. Future work can move further along three aspects:

**Query-based task interfaces.** The contrasting linear-probing trends between part segmentation and object classification suggest that a single linear probe is an overly restrictive readout for diverse downstream tasks. Moreover, the gap between linear probing and full fine-tuning on part segmentation indicates that task-relevant cues may already be present, yet are not always linearly accessible under a fixed readout. These observations point to two complementary directions for scalable adaptation. On the encoder side, a small set of global registers can be introduced to aggregate object-level semantics, providing a clean global interface for classification. On the decoder side, a task-conditioned query-based decoder can retrieve and compose fine-grained, structured information from point tokens for dense objectives such as part segmentation, with minimal changes to the pretrained encoder.

**4D spatial cognition.** While Utonia focuses on learning transferable 3D geometry from static point clouds, many real-world settings are inherently dynamic and sequence-centric. Our frame augmentation already hints at this direction by improving performance on sequence datasets such as Waymo, using pose-aligned multi-frame aggregation for the teacher and a single-frame local view for the student. Moving toward 4D spatial cognition calls for spatiotemporal pretraining objectives that enforce cross-frame consistency beyond simple aggregation. Future designs could incorporate motion-aware interactions and temporal cues, enabling representations that capture both persistent structure and evolving changes over time.

**A scalable next-generation backbone.** Scaling multi-domain pretraining, especially when coupled with task-conditioned decoders and 4D settings, will be increasingly bottlenecked by the efficiency and deployability of the sparse backbone. Sparse convolution can be memory-heavy, limiting token budget, resolution, and sequence length. Moreover, such operator stacks often introduce deployment friction due to strong kernel and system dependencies. These motivate exploring a scalable next-generation backbone that retains geometric expressiveness while being more memory- and system-friendly, favoring computation patterns that better match modern hardware and are easier to optimize and deploy at scale.

In the long run, we hope these efforts highlight sparse formats as a geometry-first interface to the physical world: compact yet expressive, naturally suited for spatial cognition, and able to complement dense videos with a scalable, queryable, and 4D-aware foundation backbone.

## Acknowledgment

This work is supported by the Hong Kong Research Grant Council General Research Fund (No. 17213925) and National Natural Science Foundation of China (No. 62422606). Here, we also thank Wenlong Huang and Jinfeng Xie for their helpful discussions.

## Impact Statement

This paper presents work whose goal is to advance the field of machine learning. There are many potential societal consequences of our work, none of which we feel must be specifically highlighted here.

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

# Appendix

*Table 7.* **Indoor and outdoor segmentation without colors or normals.** By dropping colors or colors from input data, we test Utonia's performance on modality incomplete situations, where Utonia exhibits supreme ability to adapt to such cases.

| Sem. Seg. | Params | | ScanNet w/o c. | | | ScanNet w/o n. | | | NuScenes w/o c. | | | NuScenes w/o n. | | |
|---|---|---|---|---|---|---|---|---|---|---|---|---|---|---|
| Methods | Learn. | Pct. | mIoU | mAcc | allAcc | mIoU | mAcc | allAcc | mIoU | mAcc | allAcc | mIoU | mAcc | allAcc |
| ○ PTv3 (Wu et al., 2024a) | 124.8M | 100% | 74.2 | 82.0 | 90.3 | 76.0 | 83.4 | 91.5 | 69.2 | 81.3 | 92.0 | 68.9 | 80.3 | 92.1 |
| ● Concerto (Zhang et al., 2025b) (lin.) | <0.2M | <0.2% | 36.8 | 47.3 | 73.1 | **78.2** | **87.2** | **92.1** | 42.7 | 54.3 | 81.4 | 43.6 | 55.9 | 81.5 |
| ● Utonia (lin.) | <0.2M | <0.2% | **77.0** | **86.1** | **91.5** | 77.5 | 86.9 | 91.8 | **74.5** | **83.7** | **93.6** | **75.7** | **84.3** | **93.7** |
| ● Concerto (Zhang et al., 2025b) (dec.) | 16.3M | 13% | 69.0 | 76.8 | 87.7 | 79.6 | **88.3** | 92.5 | 63.1 | 76.3 | 90.7 | 66.5 | 78.5 | 91.3 |
| ● Utonia (dec.) | 20.5M | 13% | **79.3** | **87.1** | **92.4** | **79.9** | 88.0 | **92.8** | **77.2** | **85.5** | **94.0** | **79.2** | **86.6** | **94.2** |
| ● Concerto (Zhang et al., 2025b) (f.t.) | 124.8M | 100% | 77.5 | 85.6 | 91.7 | **80.9** | **88.9** | **93.1** | 74.7 | 84.6 | 93.8 | 77.1 | 86.7 | 94.1 |
| ● Utonia (f.t.) | 157.7M | 100% | **78.5** | **87.3** | **92.3** | 80.4 | 88.7 | 93.1 | **81.1** | **87.3** | **94.6** | **82.1** | **88.4** | **94.8** |

*Table 8.* **Pretraining data**.

| Dataset | Source | Train | Val | Test | All |
|---|---|---|---|---|---|
| ScanNet (Dai et al., 2017) | real | 1,201 | 312 | 100 | 1,613 |
| ScanNet++ (Yeshwanth et al., 2023) | real | 856 | 50 | 50 | 956 |
| S3DIS (Armeni et al., 2016) | real | 204 | 68 | 0 | 272 |
| ARKitScenes (Baruch et al., 2021) | real | 4,498 | 549 | 0 | 5,047 |
| HM3D (Ramakrishnan et al., 2021) | real | 8,117 | 1,030 | 0 | 9,147 |
| Structured3D (Zheng et al., 2020) | synthesis | 16,635 | 1,722 | 1,648 | 20,005 |
| RE10k (Zhou et al., 2018b) | real | 41,670 | 0 | 4,612 | 46,282 |
| NuScenes (Caesar et al., 2020) | real | 28,130 | 6,019 | 6,008 | 40,157 |
| Waymo (Sun et al., 2020) | real | 23,691 | 5,976 | 0 | 29,667 |
| SemanticKitti (Behley et al., 2019) | real | 19,130 | 4,071 | 20,351 | 43,552 |
| HK Remote (Lands Department, 2026) | real | 6,136 | 0 | 0 | 6,136 |
| PartNet (Liu et al., 2023) | mixed | 32,537 | 0 | 0 | 32,537 |
| ScanObjectNN(Raw) (Wang et al., 2023) | real | 2,903 | 0 | 0 | 2,903 |
| GraspNet (Fang et al., 2020) | real | 6400 | 5760 | 0 | 12,160 |
| Cap3D (Luo et al., 2023) | mixed | 1,006,782 | 0 | 0 | 1,006,782 |
| Utonia | mixed | 197,868+1M | 19,797 | 32,769 | 250,434+1M |

*Table 9.* **Model Architecture.**

| Model Type | Param. | Channel Nums | Encoder Depths |
|---|---|---|---|
| Ablation | 38M | 36, 72, 144, 252, 504 | 2, 2, 2, 6, 2 |
| Main | 137M | 54, 108, 216, 432, 576 | 3, 3, 3, 12, 3 |

## A. Additional Results

In Tab. 7, Utonia shows less sensitivity to missing color or normal inputs. This supports that our Causal Modality Blinding strategy improves robustness when auxiliary channels for color and normal are absent or unreliable.

## B. Additional Implementation

### B.1. Data Preparation and Training Details

Utonia is pretrained on a diverse mixture of datasets summarized in Tab. 8. Since Cap3D is much larger than the other sources, we randomly subsample 90k Cap3D instances per training epoch to balance the mixture. Following Concerto (Zhang et al., 2025b), we apply crossmodal joint prediction using paired images when available, except for PartNet, ScanObjectNN, and HK Remote.

For outdoor datasets where point clouds do not provide color, we project image colors onto points using the provided calibration, and assign default black to points not visible in any view. Surface normals are estimated with Open3D (Zhou et al., 2018a), with directions from points to the LiDAR center. For video-based data, we reconstruct point clouds using provided camera parameters when available, like GraspNet, or feed-forward reconstruction methods (Wang et al., 2025) otherwise, like RE10K.

Joint training on various datasets can be optimization-challenging: large domain gaps and varying noise levels often increase gradient variance and lead to unstable training. We therefore adopt a two-stage pretraining schedule. In Stage 1, we pretrain on a curated set of higher-quality datasets, ScanNet (Dai et al., 2017), Structured3D (Zheng et al., 2020), Waymo (Sun et al., 2020), and PartNet (Mo et al., 2019) to obtain a stable initialization. These datasets also correspond to our ablation setting. In Stage 2, we continue pretraining for 100 epochs on the full mixture starting from the Stage 1 checkpoint. Both training stages adopt 256 as batch size and are trained on 64 NVIDIA H20 GPUs. The only difference in training parameters between

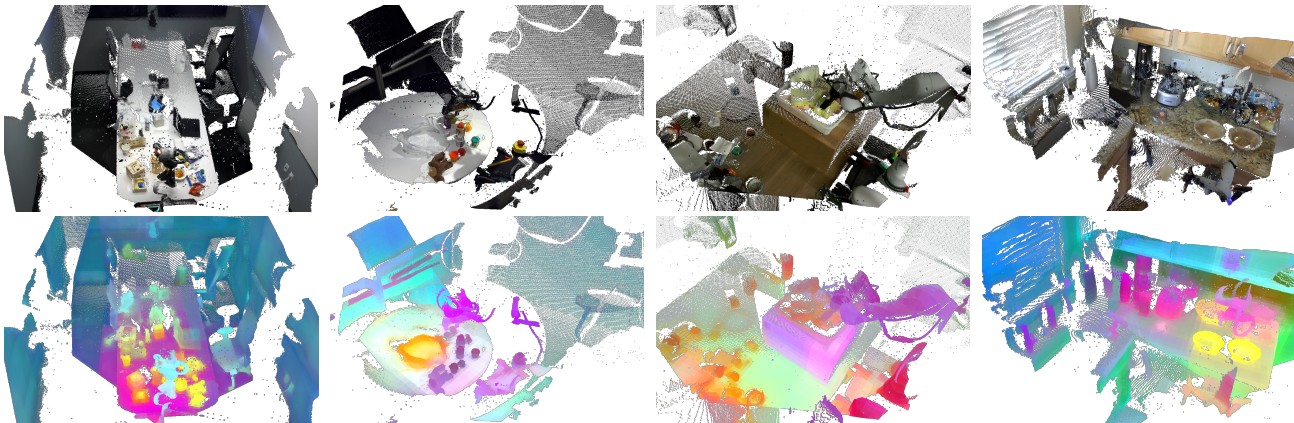

*Figure 5.* **Utonia features in cluttered manipulation scenes.** Utonia can separate objects from supporting surfaces and remain coherent under occlusion and partial observations, providing geometry-aware cues that are useful for downstream grasping and motion planning.

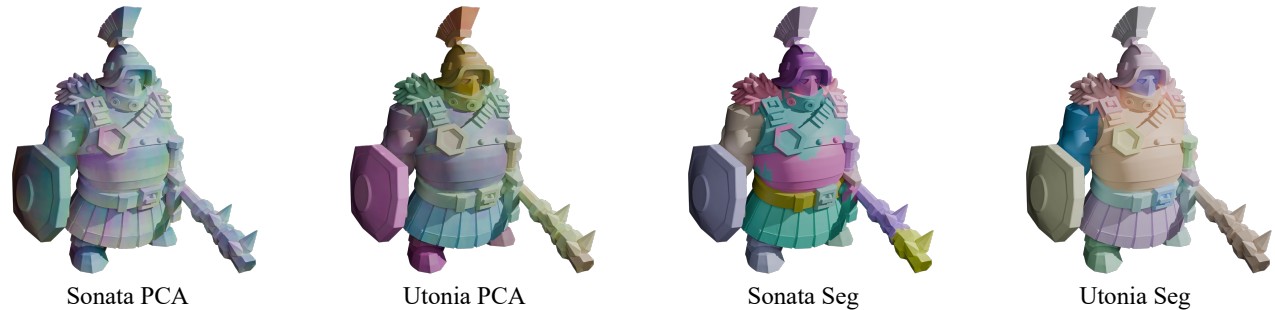

| Sonata PCA | Utonia PCA | Sonata Seg | Utonia Seg |

*Figure 6.* **Open-world object segmentation.** Utonia produces more coherent, part-aligned feature structure, leading to cleaner segment boundaries and more consistent semantic parts compared to Sonata.

Utonia and Concerto/Sonata is that the upcast level for self-distillation is set to 0 for faster training speed. Empirically, this choice of upcast level has a negligible impact on downstream performance under sufficiently large-scale pretraining. Other significant designs are provided in 3.

### B.2. RoPE Implementation Details

In 3.3, we outline the RoPE design in Point Transformer V3. Here we detail the concrete implementation. After obtaining canonical coordinates $\hat{\mathbf{p}}$, we apply coordinate jittering and scaling as in DINOv3 (Siméoni et al., 2025) to improve cross-domain generalization. Given jitter degree $\gamma > 1$, we sample an axis-wise multiplicative jitter $\mathbf{j}$:

$$\mathbf{j} = \exp(\boldsymbol{\epsilon}_j), \quad \boldsymbol{\epsilon}_j \sim \mathcal{U}\big(-\log\gamma, \ \log\gamma\big)^3, \qquad (1)$$

and apply it to the RoPE point coordinates $\hat{\mathbf{p}}_i$ as

$$\hat{\mathbf{p}}_i^j = \mathbf{j} \odot \hat{\mathbf{p}}_i, \qquad (2)$$

where $\odot$ denotes element-wise multiplication. Given scaling degree $\eta > 1$, we sample an isotropic multiplicative rescale

$$r = \exp(\epsilon_s), \quad \epsilon_s \sim \mathcal{U}\big(-\log\eta, \ \log\eta\big), \qquad (3)$$

and apply it to the previous coordinates $\hat{\mathbf{p}}_i^j$ as

$$\hat{\mathbf{p}}_i^{rj} = r\,\hat{\mathbf{p}}_i^j. \qquad (4)$$

We apply a 3D rotary positional embedding by extending the standard 1D RoPE in a separable manner across axes. Given a feature vector $\mathbf{u}$, we split it evenly into three parts $\mathbf{u} = [\mathbf{u}^x; \mathbf{u}^y; \mathbf{u}^z]$ and apply 1D RoPE to each subvector using the corresponding augmented coordinate component $\hat{\mathbf{p}}_i^{rj} = (\hat{x}_i^{rj}, \hat{y}_i^{rj}, \hat{z}_i^{rj})$:

$$\text{RoPE}_{3D}(\mathbf{u}, \hat{\mathbf{p}}_i^{rj}) = \begin{bmatrix} \text{RoPE}(\mathbf{u}^x, \hat{x}_i^{rj}) \\ \text{RoPE}(\mathbf{u}^y, \hat{y}_i^{rj}) \\ \text{RoPE}(\mathbf{u}^z, \hat{z}_i^{rj}) \end{bmatrix}. \qquad (5)$$

By applying augmented coordinates $\hat{\mathbf{p}}_i^{rj}$ to rotate queries and keys in each attention layer, this coordinate perturbation prevents the model from binding semantics to a fixed axis scaling, strengthening cross-domain transfer.

Because of the RoPE application in the attention layer, current model channels need to be divisible by 6. Here, we provide our current model settings in Tab. 9.

*Table 10.* **Open-world object part segmentation on PartObjaverse-Tiny.**

| Init | Avg. ↑ | Human-Shape | Animals | Daily-Used | Buildings | Transportations | Plants | Food | Electronics |
|---|---|---|---|---|---|---|---|---|---|
| Sonata | 55.57 | 62.54 | 61.70 | 55.23 | **42.88** | 52.21 | 63.48 | 51.63 | 54.86 |
| Utonia | **57.95** | **62.80** | **61.80** | **61.14** | 41.49 | **54.93** | **65.60** | **56.47** | **59.38** |

*Table 11.* **Robotics manipulation.** We follow the setting of GraspVLA (Deng et al., 2025) to evaluate Utonia on a simulated VLA benchmark with a monocular camera.

| Model | Sonata | Concerto | Utonia |
|---|---|---|---|
| Test Success Rate | 74.7 | 80.0 | **82.1** |

## B.3. Downstream Tasks Implementation

**Robotics manipulation.** We evaluate Utonia on table-top clutter manipulation with RGB-D observations in simulation as in (Fan et al., 2026). We synthesize a large-scale dataset from the Objaverse LVIS subset (Deitke et al., 2023), covering 290 categories and 10,680 object instances. Each episode randomly generates a physically plausible cluttered layout on a $0.4m \times 0.5m$ tabletop. Expert trajectories are generated by enumerating candidate grasp poses with BoDex (Chen et al., 2025) and planning collision-free motions with CuRobo (Sundaralingam et al., 2023), then verified for executability in MuJoCo (Todorov et al., 2012). Visual rendering is performed in Isaac Sim (Mittal et al., 2023) with extensive randomization over lighting, materials, backgrounds, and camera extrinsics. For evaluation, we construct a held-out benchmark using unseen layouts and backgrounds across 95 scenes with 15 object categories, focusing on generalization under novel clutter configurations. Depth maps are obtained either directly from the render or via depth estimation from RGB using Depth Anything V3 (Lin et al., 2025), resized to a unified resolution. For the results in Tab. 11, we report the success rate of three attempts, which is the probability of successfully grasping the object in three attempts by the gripper opening and closing. The discussions are in 4.2.

**Open-world part segmentation.** In $P^3SAM$ (Ma et al., 2025) original paper, it leverages Sonata as a 3D feature extractor and MLP-based mask decoder to do native 3D part segmentation. Although the original $P^3SAM$ employs an MLP-based mask decoder to facilitate convergence, it lacks support for flexible and interactive prompting. To address this limitation, we integrate a SAM-like decoder (Kirillov et al., 2023) architecture to enhance promptability. Under this new configuration, we observe that the Sonata pretraining proposed in the original work no longer yields satisfactory results. To evaluate the impact of different representations, we compare Sonata and Utonia as encoder initializations, performing full-parameter finetuning with identical training data and strategies. We not only present the qualitative comparisons but also further report the mIoU results of different pretrained models on PartObjaverse-Tiny (Yang et al., 2024) in Tab. 10.

*Table 12.* **Spatial reasoning.** Utonia yields the best performance. We report Acc@0.5 on ScanRefer, F1@0.5 on Multi3DRefer, CIDEr@0.5 on Scan2Cap, EM on ScanQA and SQA3D.

| Reasoning | Visual Grounding | | Captioning | Question Answering | |
|---|---|---|---|---|---|
| | ScanRefer | Multi3DRefer | Scan2Cap | ScanQA | SQA3D |
| Baseline | 51.7 | 52.7 | 83.8 | 30.1 | 58.6 |
| + Sonata | 52.6 | 52.8 | 83.5 | 30.1 | 59.7 |
| + Concerto | 52.6 | 52.7 | 79.6 | 29.6 | **60.0** |
| + Utonia | **54.0** | **54.1** | **83.9** | **30.5** | 59.9 |

**Spatial reasoning.** We build our 3D reasoning pipeline on top of Video-3D LLM (Zheng et al., 2025), which extends LLaVA-Video-7B (Zhang et al., 2025a) with a SigLIP visual encoder (Zhai et al., 2023) and a Qwen2-7B language model (Team et al., 2024). For each image patch, we compute the 3D coordinates of its patch center to form a pseudo point cloud, and introduce Utonia as an additional point cloud encoder to extract geometry-aware features. We fuse these 3D features with the original 2D visual features by addition, injecting explicit geometric cues into the visual tokens. All training settings follow the default configuration of Video-3D LLM (Zheng et al., 2025), and Utonia is kept frozen. The evaluation results are provided in Tab. 12.

