# OpenReview forum: "Utonia: Toward One Encoder for All Point Clouds"
_ICML.cc/2026/Conference — ICML 2026 regular_

### Official Review · Reviewer_n1q1 · 2026-02-19

**Soundness:** 4
**Presentation:** 4
**Significance:** 4
**Originality:** 4
**Overall Recommendation:** 5
**Confidence:** 4

**Summary:**

This paper presents a unified 3D point cloud encoder designed to generalize across various 3D data source. The paper argues that it is domain-specific priors instead of inherent geometry difference that drives fragmentation in 3D modeling. Built on an enhanced Point Transformer, the proposed method achieves domain-agnostic representation learning. Pretrained on large-scale mixed data, it achieves competitive performance across benchmarks and shows the Utonia representations can benefit embodied and multimodal reasoning.

**Compliance With Llm Reviewing Policy:**

Affirmed.

**Final Justification:**

The rebuttal directly address my concerns, particularly regarding the 3D RoPE and the concern of catastrophic forgetting. Overall, the insightful clarifications solidify my recommendation for acceptance of this technically strong paper.

**Key Questions For Authors:**

- How does the 3D RoPE implementation handle the quantization noise inherent in sparse grid-based transformers, particularly when coordinate jittering is applied to tiny RoPE bases?.

- The results show that PartNetE linear probing drops with Utonia, but fine-tuning improves well over baselines. Does this imply a semantic shift in the pretrained representation space that favors global scene understanding over local part geometry?

- Given the two-stage training schedule, did the authors observe catastrophic forgetting of Stage 1 datasets during Stage 2, or did the balanced subsampling of Cap3D effectively regularize the full mixture?

**Limitations:**

yes

**Strengths And Weaknesses:**

**Strengths**

* The authors have succeeded in breaking the long-standing gaps between indoor, outdoor, and object-centric pretraining. By clearly identifying the mismatches that usually trip up these models, specifically scale, gravity, and modality, they’ve laid out a highly practical roadmap for scaling 3D foundation models.


* Bringing 3D Rotary Positional Embeddings (RoPE) into the transformer architecture is a valuable technical contribution. It gives the model a parameter-free way to handle wild variations in density and scale without losing relative spatial relationships.


* The enlarged scale of this pretraining is huge. It clearly pays off, as the features generalize well to downstream tasks like robotic manipulation and multimodal spatial reasoning.


* Utonia is highly robust to missing data, holding up remarkably well even when color or surface normal channels are dropped. To keep the joint optimization of such wildly diverse datasets from collapsing, the authors propose a two-stage training schedule and rescaling to stabilize the whole process.

**Weaknesses**

* Implementing 3D RoPE forces a structural constraint where channel dimensions absolutely must be divisible by 6. While manageable, it restricts certain architectural tweaks for hardware-specific optimizations.


* The reliance on a carefully-crafted, two-stage pretraining process proves that optimizing city-scale LiDAR alongside unit-sphere objects from scratch is still challenging. It may reflect that the multi-domain 3D loss landscape remains prone to instability.


* While Utonia address global semantic understanding, getting strong part-level segmentation relies on fine-tuning rather than the raw pretrained features. This may suggest the self-distillation setup favors high-level semantic invariance over preserving fine-grained geometric details.

---

> ### Author Rebuttal · Authors · 2026-03-31
>
> Dear Reviewer n1q1,
>
> We truly appreciate your close reading of Utonia. It is especially encouraging to see your recognition of both the scale of the unified pretraining effort and the robustness of the resulting representation, particularly under missing modalities and across a broad range of downstream tasks.
>
> ---
>
> ## R1: RoPE under Positional Noise
>
> Thank you for this very good question. In our design, 3D RoPE is intended only to describe relative positional relationships within PTv3's local attention region, rather than across the whole scene. Therefore, the RoPE base only needs to cover the local attention span after rescaling. In practice, we choose the RoPE base along with the local attention range, rather than treating it as a standalone hyperparameter.
>
> Regarding the quantization noise introduced by sparse discretization or coordinate jittering, our view is that it is not handled by RoPE alone. The jitter we apply is moderate, and the local attention mechanism naturally absorbs small perturbations through neighborhood aggregation. In this sense, RoPE provides the relative positional signal, while attention handles the small positional noise in practice.
>
> ---
>
> ## R2: Interpreting the PartNetE Linear-Probing Gap
>
> This is a very good question. We do not think the lower PartNetE linear-probing result means that Utonia gives up local part geometry in favor of global scene semantics. A more likely explanation is that unified pretraining changes the organization of the representation, rather than erasing the part-aware information itself.
>
> The key evidence is that full fine-tuning improves clearly over baselines. If local part geometry had really been lost, we would expect fine-tuning to also suffer, but this is not what we observe. Instead, our reading is that fine-grained part cues are still present, but they are no longer equally exposed to a fixed linear probe.
>
> So in our view, this gap says more about the limitation of linear probing than about a deficiency of the representation itself. We also see this as an important future direction: building stronger task-conditioned or query-based readout mechanisms that can better recover structured local information from a shared latent space (let’s say, a bit more JEPA-like).
>
> ---
>
> ## R3: Why No Catastrophic Forgetting
>
> Thanks for pointing out this concern. Our two-stage schedule is not designed as a domain-shifting fine-tuning phase, but rather as a clean-to-scale strategy. In Stage 1, we train on a cleaner mixture so that the model can first build a stable basic geometric understanding. Stage 2 then scales this up, but it still covers the Stage 1 data rather than replacing it.
>
> Therefore, under sufficient model capacity, Stage 2 is not really a forgetting phase, but a continuation of joint pretraining on a broader mixture. In addition, the largest source, Cap3D, is balanced through subsampling, so it does not overwhelm the other domains. Empirically, we do not observe clear signs of catastrophic forgetting. If the Stage 1 datasets had been substantially forgotten during Stage 2, we would expect their downstream performance to collapse, which is not what we observe.
>
> Best,
> Utonia Authors

---

> > ### Author Rebuttal · Reviewer_n1q1 · 2026-04-01
> >
> > Thank you for the clear and comprehensive response. It directly address my concerns, particularly regarding the 3D RoPE and the concern of catastrophic forgetting. Overall, your insightful clarifications solidify my recommendation for acceptance of this technically strong paper.

---

> > > ### Author Response · Authors · 2026-04-01
> > >
> > > Dear Reviewer n1q1,
> > >
> > > Thank you again for recognizing our research. We want to assure you that our commitment to this direction will not end with Utonia. We will continue to encourage and support future research, and help push toward a better future for 3D point clouds and sparse representations.
> > >
> > > Best,
> > > Utonia authors

---

### Official Review · Reviewer_pxTa · 2026-03-06

**Soundness:** 2
**Presentation:** 2
**Significance:** 3
**Originality:** 3
**Overall Recommendation:** 3
**Confidence:** 5

**Summary:**

This paper proposes to train a unified point cloud encoder across heterogeneous domains. By identifying three key inconsistencies that hinder cross-domain unification: (1) input channels, (2) sparsity and density, and (3) granularity, the authors proposed three simple yet effective designs: (1) causal blinding, (2) RoPE-enhanced positional hints, and (3) granularity-prompted rescaling. Experiments across object-, indoor-, and outdoor-scale, and also spatial reasoning and robotic tasks, have supported the effectiveness of the pretraining.

**Compliance With Llm Reviewing Policy:**

Affirmed.

**Final Justification:**

I note that in the response, both reviewers n1q1 and qvxH acknowledge the issues I raised. However, neither provides a concrete explanation grounded in empirical evidence, theoretical analysis, or controlled experiments. I therefore do not find that reframing the contribution or relying on qualitative arguments sufficiently addresses these concerns. While I appreciate the authors’ effort to tackle an important problem, I still believe the paper does not meet the standard required for acceptance at ICML.

**Key Questions For Authors:**

1. Can you show more support for the claim that “object-, indoor-, and outdoor-scale data begin to mutually improve one another under a shared encoder, rather than compete across domains”?
2. What is the benefit of such gravity-irrelevant representations?

Either experimental or theoretical support could strongly improve this paper.

**Limitations:**

No. It is suggested to include a limitations discussion.

**Strengths And Weaknesses:**

Strengths:
1. This research holds significant practical relevance. The domain fragmentation in point cloud data has hindered large-scale training for a long time; this paper addresses this problem with an effective solution.
2. The pilot study is solid and inspiring, and the emergent behaviors of Utonia (such as insensitivity to gravity priors and consistency across different domains) are valuable.

Weaknesses:
1. Despite the improvement in the downstream tasks, the novelty of the methodology is somewhat weak. The three core contributions: coordinate rescaling, 3D RoPE, and modal Dropout, are all combinations of existing techniques, presenting more as an engineering experience summary.
2. Such a design could be acceptable with proper theoretical justification; however, the paper lacks experimental and theoretical evidence showing that the gains from joint pretraining are attributable to these design choices.
3. The presentation of the paper is somewhat unsatisfactory. The method lacks structural clarity. The difference between the upper and lower parts of Figure 4 is unclear. The details of the training method are lacking (now only the proposed augmentation design). In Section 4.1 Main Results, two paragraphs appear to be duplicated. Moreover, the paper lacks a conclusion section. The authors should carefully proofread and revise the manuscript.

---

> ### Author Rebuttal · Authors · 2026-03-31
>
> Dear Reviewer pxTa,
>
> Thanks for the candid feedback. We are especially encouraged that, despite your concerns, you clearly recognized the practical importance of this problem, the value of the pilot study, and the emergent behaviors shown by Utonia. To us, this already touches the core motivation behind the work.
>
> More broadly, we see the goal of Utonia as part of a larger dream that has been gradually shared across the point cloud community for years: point clouds from different domains should not have to learn in isolation forever. Indoor scenes, outdoor LiDAR, remote sensing, object-centric scans, and video-lifted point clouds have long evolved as fragmented directions, often each building its own training recipe, benchmark habit, and representation prior. If these domains can begin to benefit from a shared pretrained encoder, then progress in foundation-model-style point representation learning no longer needs to radiate into each domain separately. That possibility itself is already important.
>
> From this perspective, we do not see simplicity as a weakness. On the contrary, there is something especially meaningful when a simple yet effective recipe can begin to unlock such a long-standing and community-shared goal. Utonia is certainly not created in isolation. It stands on the collective efforts of many researchers who have pushed point encoders, self-supervised learning, multimodal learning, and large-scale pretraining forward over the years. Precisely because of this accumulated progress, it becomes possible for a relatively clean and elegant design to produce surprisingly broad gains. To us, that is not merely an engineering coincidence, but a beautiful sign that this direction is becoming mature enough to unify.
>
> Starting from Utonia, we hope the field can more seriously explore a future in which advances in pretrained 3D representation learning are shared across domains, rather than each domain continuing to fight alone. We therefore deeply appreciate your engagement with both the strengths and the limitations of the current version, and below we clarify the specific evidence and presentation issues more explicitly.
>
> Regarding your further concern related to the design effectiveness, we wish our following response will solve your problem:
>
> ---
>
> **R1: Design Effectiveness**
>
> - Causal Modality Blinding. In **Table 12** of the Appendix, we report indoor and outdoor semantic segmentation without color or normal. For object-centric data, ScanNetObjectNN and ModelNet40 do not have colors, and the results are shown in **Table 5**. These results show that Utonia remains robust when colors or normals are absent, whereas Concerto is not.
>
> - Perceptual Granularity Prompted Rescaling. In **Table 2**, the results show that compared with applying domain-specific grid sizes and grid-size jitter, perceptual granularity rescaling aligns spatial units, leading to performance closer to previous single-domain results.
>
> - 3D RoPE for Cross-Domain Spatial Encoding. In **Table 6**, the results show that RoPE brings limited gains in single-domain pretraining, but yields consistent improvements under multi-domain training, proving that RoPE helps spatial interaction in cross-domain training.
>
> ---
>
> **R2: Mutual Benefit Across Domains**
>
> **Table 2** shows that naively mixing cross-domain data leads to poor performance, while granularity rescaling substantially reduces the gap between joint training and separate-domain training. **Table 6** further shows that RoPE brings limited or inconsistent gains under single-domain pretraining, but consistent improvements under multi-domain pretraining, suggesting that different domains begin to benefit from shared pretraining rather than merely compete. This is further supported by **Tables 3, 4, and 5**, where a single Utonia encoder matches or outperforms pretrained domain-specific models across different domains separately.
>
> ---
>
> **R3: Benefits of Gravity-Irrelevant Representations**
>
> For object-centric data, gravity-irrelevant representations are necessary. As in Figure 3, the flowers should not exhibit large different representation patterns when placed in different directions. Moreover, we do not seek to remove gravity completely. Our view is that gravity relevance should be granularity-dependent. At scene scale, gravity is still a meaningful prior for layout and upright-world understanding. But at object scale, gravity dependence can hurt intrinsic geometric understanding.
>
> ---
>
> **R4: Enhancing Details**
>
> Thanks for your suggestions. The preliminary method and conclusion will be included in the final version. Due to space limitations, we do not attach here. The difference between the upper and lower parts of Figure 4 is as in the caption: cross-domain data, RoPE-enhanced Point Transformer V3 with augmentations for domain-prior erasing, broader evaluations extending segmentation to spatial reasoning, robotic manipulation, and open-world part segmentation.
>
> Best,
> Utonia Authors

---

> > ### Author Rebuttal · Reviewer_pxTa · 2026-04-04
> >
> > I understand the authors’ ambitious vision and expectations for this work; however, I do not believe this justifies overlooking the methodological rigor and theoretical soundness of the approach.
> >
> > First, in my understanding, “mutually improve one another” should imply that training on one domain yields benefits for the other. For example, training solely on indoor data should improve performance on outdoor data, or joint training on indoor and outdoor data should outperform training on either domain alone. I was unable to find evidence supporting this claim in Table 2 and Table 6; on the contrary, the results suggest that joint training does not significantly outperform single-domain training.
> >
> > Second, as noted in my previous comments, the authors do not clearly explain the relationships among the three main components. These components appear more akin to heuristic adjustments rather than a well-motivated, coherent design.
> >
> > Based on these points, I do not consider my primary concerns to have been adequately addressed.

---

### Official Review · Reviewer_wiC6 · 2026-03-06

**Soundness:** 4
**Presentation:** 3
**Significance:** 3
**Originality:** 3
**Overall Recommendation:** 5
**Confidence:** 2

**Summary:**

This paper proposes a unified self-supervised point transformer encoder, Utonia, which can be jointly trained across multiple point cloud domains, including indoor RGB-D scenes, outdoor LiDAR scans, object-centric CAD models, and point clouds lifted from RGB-only videos. To address these challenges, three domain-agnostic design components are introduced: granularity-prompted rescaling based on canonical coordinates to harmonize perception scales; 3D RoPE-based spatial encoding for consistent positional representation across domains; and causal modality blinding to improve robustness to missing color or normal channels. The model is pre-trained on a large multi-domain hybrid dataset and evaluated on downstream tasks including indoor and outdoor semantic segmentation, object classification and part segmentation, robotic manipulation, and 3D spatial reasoning. Results demonstrate that unified cross-domain pre-training is feasible and yields relatively state-of-the-art transfer performance compared to previous domain-specific training methods.

**Compliance With Llm Reviewing Policy:**

Affirmed.

**Final Justification:**

The authors have provided satisfactory responses to my questions, and I consider my concerns to be resolved.

While the rebuttal improves my understanding of the work, it does not substantially change my overall evaluation of its contribution and significance, so I keep my original score.

**Key Questions For Authors:**

1.The conclusions of the paper largely depend on the stability and effectiveness of large-scale cross-domain joint pretraining. Can the authors provide a clearer description of the data mixture proportion and sampling strategy of the multi-domain datasets? If one domain accounts for too much or too little in the sampling, will it have a significant impact on the results? The answer of the authors will affect the evaluation of the reliability of the paper.

2.Is the choice of the rescaling factor and the base granularity in this paper derived from some interpretable principle or theory, or is it mainly determined through empirical parameter tuning? If the authors can provide clear reusable rules, the reliability and reproducibility of the paper will be improved.

3.Will the training configuration, code, or pretrained weights be released?

**Limitations:**

yes

**Strengths And Weaknesses:**

Soundness:
The paper is generally reasonable, logical, and self-consistent. The empirical support is relatively sufficient: not only multi-task evaluations, but also ablation studies explain the contribution of each component The authors are also honest about the boundaries of their work.

Presentation:
The structure and narrative thread of the paper are relatively clear. However, the key training framework relies heavily on citations in this paper. The authors repeatedly ask readers to directly refer to the original papers, which is not friendly to readers who first encounter this direction and also raises the threshold for understanding. Compared with existing methods, the paper mainly lists the differences and verifies the effectiveness through experiments, while the analysis of why these differences can systematically eliminate domain shortcuts or domain priors and promote cross-domain alignment is still limited. If the core recipe could be presented in a more concise pseudocode or table, the readability and reproducibility of the paper would be stronger.

Significance:
This paper gives a clear solution to the problem of cross-domain unified point cloud representation, which is one of the core bottlenecks of 3D representation learning. It has certain research and application value: a unified point encoder across domains can reduce the cost of repeated training and maintenance for downstream tasks in different domains; also, this paper decomposes the phenomenon that domain priors lead to representations clustering by domain into three types of operable mismatches and gives corresponding engineering fixes, which are easy to be inherited or extended by subsequent work.
However, its influence is also limited by the threshold: the cross-domain joint pretraining require high data and model scale, and the effect may be sensitive to the data mixture strategy and scale, which will increase the difficulty of practical application and independent reproducibility.

Originality:
The contribution of the paper is mainly mechanism interpretation and design integration: instead of introducing a new theory, it explains the cause of the failure of cross-domain joint pretraining, and accordingly reorganizes several existing ideas into a set of minimal-change recipes that are more suitable for cross-domain point clouds. Although a single component may not be completely new, the pipeline of problem decomposition – design correspondence – cross-task evaluation forms a relatively complete closed loop. In particular, perceptual granularity prompted rescale is taken as the entry point for scale alignment, and it is further integrated into the construction of patial encoding, showing clear originality.

---

> ### Author Rebuttal · Authors · 2026-03-31
>
> Dear Reviewer wiC6
>
> Thanks for your encouraging feedback and for recognizing the closed-loop pipeline behind Utonia, from problem decomposition to design correspondence to cross-task evaluation. We especially appreciate that you recognized the work not just as a collection of empirical results, but as a more complete research loop connecting diagnosis, method design, and validation across diverse settings.
>
> ---
>
> ## R1: Open-Source Commitment
> As student researchers, we have been deeply shaped by the open-source culture of our community. We are therefore fully committed to open-sourcing Utonia, including the training configuration, codebase, pretrained weights, inference guidance, and online demos. We will also continue responding to questions from the community, and hope Utonia can benefit the community in the same way that we have benefited from it.
>
> ---
>
> ## R2: Data Mixture and Sampling Strategy
>
> Thank you for raising this important question. Since the reliability of Utonia indeed depends on stable large-scale cross-domain joint pretraining, we attach the relevant details on the data mixture and sampling strategy as follows.
>
> A key point in our setting is that balancing multi-domain point clouds is not purely a sample-level issue, since the effective training signal is more directly governed by the number of points contributing to the loss. For example, one object-level sample typically contains far fewer points than one large scene-scale scan. This is also why a seemingly balanced sample count may still lead to an imbalanced pretraining signal across domains.
>
> In our current recipe, each dataset is traversed once in turn during an epoch. The only exception is the largest source, the 1M Cap3D set, which is randomly subsampled to 90,000 objects per epoch. The dataset composition is summarized in Table 11, and we will make this sampling rule explicit in the final version.
>
> We intentionally keep this recipe simple. Our goal is not to carefully tune a fixed mixture ratio for the best benchmark performance, but to test whether a single encoder can benefit from broader cross-domain data under a general and reproducible setup. It is true that if one domain is oversampled or undersampled, it can indeed affect the final behavior. Therefore, in our current setting, we cap the largest object source, to avoid obvious domination from a single domain. A more systematic study of mixture sensitivity is also an important future direction that we would be glad to discuss more clearly in the final version.
>
> For completeness, we will also add the corresponding pretraining configuration in the final version, including batch size, image preprocessing, optimizer, scheduler, and other implementation details. The full pretraining code will also be released.
>
> ___
>
> ## R3: Selection Principles for Granularity Rescaling Factor
> The choice is not from a strict closed-form theory, but it is also not free tuning. In our practice, it is mainly guided by two simple principles: (1) after rescaling, the point cloud should still remain well connected under the grid-based local operator, rather than becoming too sparse or fragmented; and (2) the granularity should match the perceptual scale we want the encoder to work at. For example, object-centric point clouds and outdoor scenes should clearly not be treated at the same granularity.
>
> In practice, we start from these two principles and then make light experiment-driven adjustments, rather than arbitrarily tuning for the best benchmark number. In this sense, Utonia is not introducing another arbitrary scale, but turning previously separate domain-specific choices into a more unified granularity-alignment recipe across domains. We will make these reusable rules explicit in the final version.
>
> ___
>
> ## R4: Pipeline Pseudocode
> We agree that clean pseudocode would help readers better understand the overall pipeline. We will include it in the appendix of the final version. Due to the response length limit, we are unable to attach it directly here.
>
> Best,
> Utonia Authors

---

> > ### Author Rebuttal · Reviewer_wiC6 · 2026-04-03
> >
> > The authors have provided satisfactory responses to my questions, and I consider my concerns to be resolved.
> >
> > While the rebuttal improves my understanding of the work, it does not substantially change my overall evaluation of its contribution and significance, so I keep my original score.

---

> > > ### Author Response · Authors · 2026-04-05
> > >
> > > Thank you again for recognizing the value of our work and for your constructive suggestions, which helped us better understand how to improve the paper. We will continue putting effort into the open-source release of Utonia, including code, model weights, and supporting materials, and do our best to keep engaging with the community so that Utonia can be useful beyond the paper itself.
> > >
> > > Best,
> > > Utonia Authors

---

### Official Review · Reviewer_qvxH · 2026-03-12

**Soundness:** 3
**Presentation:** 4
**Significance:** 4
**Originality:** 3
**Overall Recommendation:** 5
**Confidence:** 4

**Summary:**

Utonia addresses a core challenge in 3D perception: point cloud self-supervised learning (SSL) has remained domain-fragmented, with separate models for indoor, outdoor, object-centric data, etc. The paper proposes a single unified point transformer encoder pretrained across all these domains simultaneously with techniques addressing robustness and generalization.
The authors identify three key obstacles to joint training: scale/density mismatches, gravity-prior bias (scene models rely on z-up orientation), and inconsistent modality availability (color, normals). They address these with three targeted, domain-agnostic fixes: Perceptual Granularity Prompted Rescaling (canonicalizing coordinate scales), 3D RoPE (rotary positional embeddings for transferable spatial encoding), and Causal Modality Blinding (random dropout of color/normal channels during pretraining). The backbone is Point Transformer V3, extended to V3.5 with these modifications.
Trained on ~250k cross-domain scenes plus 1M CAD objects, Utonia matches or surpasses the prior SOTA (Concerto, Sonata) across indoor, outdoor, and object benchmarks, and additionally transfers to robotics manipulation and VLM spatial reasoning tasks.

**Compliance With Llm Reviewing Policy:**

Affirmed.

**Final Justification:**

My concerns are address, I maintain my opinion to accept this paper.

**Key Questions For Authors:**

The motivation is clear and the approach is solid, but some presentation and clarity issues prevent a higher confidence rating, leading to a rating of "Accept" for now.

1. The paper falls short of the 8-page limit. Could the authors use the remaining space to (a) provide a self-contained explanation of the self-distillation scheme rather than deferring entirely to Sonata, and (b) elaborate on the downstream experiments, particularly the robotics manipulation setup which goes unreferenced in the main text, and the feature fusion strategy used in the spatial reasoning experiments?
2. Why are Sonata's outdoor segmentation results reported with additional PPT supervision and extra data when all other methods are evaluated without these advantages? Could the authors either provide a fair comparison or justify this choice? It would also help to briefly explain what PPT is for readers unfamiliar with it.
3. What exactly is Table 2 showing? The text in Section 2 refers to it as a Concerto evaluation, but it is presented as motivation for Utonia's design choices. Are these results from Concerto, Utonia, or some intermediate model? Clarifying this is important for interpreting the pilot study correctly.

**Limitations:**

Some discussion would be appreciated.

**Strengths And Weaknesses:**

S1. The motivation and pilot study are highly convincing. Rather than simply asserting that domain gaps are problematic (which they are), the authors isolate and quantify three concrete failure modes (scale/density mismatch, gravity-prior leakage, and modality inconsistency) and demonstrate empirically in Table 2 that naive fixes like grid-size jittering fall short. This kind of honest diagnostic groundwork makes the problem setup feel earned rather than assumed.

S2. The proposed techniques are simple but well-motivated, with each design choice tied directly to a specific failure mode identified in the pilot study. None of them feel like solutions in search of a problem. The quantitative results across indoor, outdoor, and object benchmarks are competitive with or better than domain-specific baselines, which is a non-trivial bar to clear given the generalist nature of the model.

S3. The ablation studies are very thorough and intellectually honest. The authors are willing to report cases where a design choice offers marginal or mixed benefit (e.g., RoPE coordinate augmentation), which builds trust in the reported gains elsewhere. The downstream evaluations on robotic manipulation and VLM spatial reasoning are a nice addition, extending the paper beyond the standard segmentation benchmarking, although I would like to see a little bit more extensive evaluation and clearer explanation of the experimental settings.

S4. The qualitative figures are among the strongest aspects of the paper. Figures 1, 2, 3, 5, and 6 do real argumentative work in addition to being visually pleasing. The cross-domain PCA visualizations and the toy-car semantic similarity comparison in Figure 2 are especially effective at building intuition for what the unified representation is actually capturing.

W1. The paper falls short of the 8-page limit, which suggests either a lack of content or insufficient polish. The space could have been utilized in a couple of ways (in fact, any way is better than leaving it empty):
- The self-distillation scheme is never properly explained, with readers simply pointed to Sonata. This is unsatisfying given that Sonata is a separate, previously published work and the mechanism is not canonical enough to be glossed over. A concise description within the paper itself is warranted.
- The downstream application experiments receive surprisingly thin treatment. The robotics manipulation experiment is not even referenced in the main text body, and the spatial reasoning section leaves a key question unanswered: how exactly are the point encoder features fused into the video-based 3D LLM pipeline? A few sentences of clarification would go a long way.

W2. In the outdoor semantic segmentation results (Table 4), Sonata is reported with additional supervised tuning via PPT and extra data, making it not directly comparable to Utonia. This is acknowledged but not justified and it is unclear to me the need to make this an unfair comparison? Additionally, even a brief explanation of PPT would be helpful.

W3. It is unclear what exactly Table 2 is showing. The text in Section 2 describes it as evaluating Concerto under different grid size settings, yet it is presented in the context of motivating Utonia's design. Clarifying whether these are Concerto or Utonia results is important for interpreting the pilot study.

W4. On the topic of polish, Section 3.4 header reads "Causal modality blinding." while all other section headers follow title case (capitalized). A minor issue, but one that could reflect a lack of proofreading attention in the writing process.

---

> ### Author Rebuttal · Authors · 2026-03-31
>
> Dear Reviewer qvxH,
>
> Thanks for the encouraging feedback and for recognizing Utonia as a simple yet effective practical roadmap toward one encoder for all point clouds. We particularly appreciate your view that the contribution of Utonia lies not only in the final performance and broad downstream results in VLM reasoning and robotics, but also in the diagnostic process and the problem setup itself. We have always believed that there is a special elegance in using simple ideas to address an important challenge, and we are glad to see this value resonate with you.
>
> ---
>
> ## R1: Enhancing Details
>
> We absolutely agree that the remaining page budget can be put to better use, and we would be happy to use it to make Utonia an even better paper. We truly appreciate these detailed suggestions, and are glad to make Utonia even better together. In the final version, we will add the following:
>
> - A short preliminary section at the beginning of the methodology to briefly introduce the ideas inherited from Sonata and Concerto;
> - More implementation details for the VLA-based robotics manipulation setting (refer to Any3D-VLA) and the VLM-based spatial reasoning pipeline (refer to Video-3D LLM) in the Appendix;
> - A concluding discussion on future directions and broader impact to encourage follow-up research.
> - Consistent headers for subsections.
>
> ```
> Robotics Manipulation: the datasets and evaluation settings are listed in Appendix B.2. For the method, we leverage Utonia as the 3D encoder for lifted point clouds, finetuning only the last few sparse convolution layers. The point cloud features are then concatenated with 2D features from DINOv2 and SigLIP to obtain a residual vector for image features. The fused representations, together with the language tokens, are fed into a VLA backbone. In Table 8, we report the success rate across three attempts, demonstrating Utonia's progress in object-centric robotics tasks.
>
> VLM Spatial Reasoning: For each image patch, we compute the 3D coordinates of its center to form a pseudo-point cloud and introduce Utonia as an additional point-cloud encoder to extract geometry-aware features. We fuse these 3D features with the original 2D visual features by addition, injecting explicit geometric cues into the visual tokens. All training settings follow the default configuration of Video-3D LLM, and Utonia is kept frozen. The evaluation results are provided in Table 9.
> ```
>
> ---
>
> ## R2: Why Sonata+PPT vs. Pure Utonia
>
> This is a very detailed and important question. We report Sonata+PPT simply to stay faithful to the original Sonata paper. It is not meant to be a strictly apples-to-apples comparison to pure Utonia, but rather a record of what that earlier stage of this research line achieved.
>
> At that stage, PPT was a useful paradigm for addressing domain gaps and supported works like PTv3, and Sonata in studying joint training at a larger scale. With Utonia, however, we now see a cleaner and more elegant way to train a self-supervised model across domains directly, and PPT has already done its job for us. For this reason, we intentionally keep Utonia clean, without adding PPT on top.
>
> More broadly, we hope this also encourages the community to move beyond earlier dataset-specific fixes and seek a more unified joint training recipe for general 3D perception. We will also add a brief note in the final version to make the role of PPT more explicit.
>
> ---
>
> ## R3: Concerto Pilot Study vs. Utonia Motivation
>
> Table 2 reports Concerto results, rather than Utonia. We include it as a pilot study because it provides early evidence for the core issue that later motivates Utonia: simply using domain-specific grid sizes, or adding grid-size jitter, is still not enough to resolve the scale and density mismatch across domains.
>
> More specifically, grid size sets the metric unit of local neighborhoods, so the same operator may cover centimeters in one domain but meters in another. Such granularity shifts change neighborhood structure and make the learned features sensitive to domain-specific cues. This is exactly why we later move to Utonia: instead of adapting to each domain separately, we rescale each point cloud to a standard observing granularity, so that positional hints and local interactions are built upon more comparable spatial units across domains.
>
> We will make this distinction more explicit in the final version, so that Table 2 is understood as a motivating Concerto-based pilot study, rather than a Utonia result.
>
> Best,
> Utonia Authors

---

> > ### Author Rebuttal · Reviewer_qvxH · 2026-04-04
> >
> > My concerns have been addressed.

---

> > > ### Author Response · Authors · 2026-04-05
> > >
> > > It means a lot to us that the ideas behind Utonia resonated with you! We are equally thankful for your detailed suggestions, which gave us a clearer sense of how to strengthen the paper. We will keep improving the final version and continue working on the open-source release of Utonia, in the hope that it can grow beyond the paper itself into something the community can genuinely use, reproduce, build upon, and push further.
> > >
> > > Best,
> > > Utonia Authors

---

### Decision · Program_Chairs · 2026-04-30

**Decision:**

Accept (regular)

**Comment:**

The final ratings are 3 Accepts (qvxH, wiC6, n1q1) and 1 Weak Reject (pxTa). The AC has read the reviews and the rebuttal.

The reviewers appreciated the motivation of pilot study and proposed method (qvxH, pxTa, n1q1) and thorough ablation studies (qvxH, wiC6).
However, the reviewers also raised a number of concerns, including
lack of description in training framework and self-distillation scheme (qvxH, wiC6, pxTa), insufficient experiments (qvxH), inconsistent training schemes between methods compared in Table 4 (qvxH), lack of novelty and primarily applies existing components (wiC6, pxTa), lack of justification (pxTa), poor presentation (pxTa), dimensions of channels must be divisible by 6 (n1q1), reliance on carefully crafted two-stage pretraining (n1q1). Note that pxTa raised several critical weaknesses regarding the claims being not well-supported by empirical evidence, and that overall there is a lack of rigor and justification in the methodology. Reviewers n1q1 and qvxH discussed and maintained their scores; pxTa did not follow up on their comment.

Overall, the recommendation is leaning towards acceptance. Nonetheless, the AC recognizes the points raised by pxTa. The AC recommends the authors to consider the feedback and incorporate the materials presented in the rebuttal, which would improve the next revision of the manuscript.